# Cluster randomised controlled trial to investigate the effectiveness and cost-effectiveness of a Structured Health Intervention For Truckers (the SHIFT study): a study protocol

Stacy A Clemes [1], Verónica Varela Mato,[1] Fehmidah Munir,[1] Charlotte L Edwardson,[2] Yu-Ling Chen,[1] Mark Hamer,[1] Laura J Gray,[3] Nishal Bhupendra Jaicim,[4] Gerry Richardson,[5] Vicki Johnson,[6] Jacqui Troughton,[6] Thomas Yates,[2] James A King[1]

For numbered affiliations see end of article.

**Correspondence to**
Dr Stacy A Clemes;
s.a.clemes@lboro.ac.uk

## ABSTRACT

**Introduction** Heavy goods vehicle (HGV) drivers exhibit higher than nationally representative rates of obesity, and obesity-related comorbidities, in comparison to other occupational groups. Their working environments are not conducive to a healthy lifestyle, yet there has been limited attention to health promotion efforts. We have developed a Structured Health Intervention For Truckers (the SHIFT programme), a multicomponent, theory-driven, health-behaviour intervention targeting physical activity, diet and sitting in HGV drivers. This paper describes the protocol of a cluster randomised controlled trial designed to evaluate the effectiveness and cost-effectiveness of the SHIFT programme.

**Methods and analysis** HGV drivers will be recruited from a logistics company in the UK. Following baseline measurements, depots (clusters) will be randomised to either the SHIFT intervention or usual-care control arm (12 clusters in each, average cluster size 14 drivers). The 6-month SHIFT intervention includes a group-based interactive 6-hour education session, worksite champion support and equipment provision (including a Fitbit and resistance bands/balls to facilitate a 'cab workout'). Objectively measured total daily physical activity (steps/day) will be the primary outcome. Secondary outcomes include: objectively measured light-intensity physical activity and moderate-to-vigorous physical activity, sitting time, sleep quality, markers of adiposity, blood pressure and capillary blood markers (glycated haemoglobin, low-density lipoprotein-cholesterol, high-density lipoprotein-cholesterol). Self-report questionnaires will examine fruit and vegetable intake, psychosocial and work outcomes and mental health. Quality of life and resources used (eg, general practitioner visits) will also be assessed. Measures will be collected at baseline, 6 and 12 months and analysed according to a modified intention-to-treat principle. A full process evaluation and cost-effectiveness analysis will be conducted.

**Ethics and dissemination** Ethical approval was obtained from the Loughborough University Ethics Approvals Sub-Committee (reference: R17-P063). Study findings will

### Strengths and limitations of this study

► To our knowledge, this is the first randomised controlled trial to examine the impact of a multi-component intervention targeting individual and environmental barriers faced by heavy goods vehicle drivers to lead a healthy lifestyle.
► The trial will involve a full process evaluation and an economic evaluation.
► The primary outcome, physical activity, will be objectively measured.
► Sustainability of the intervention will be examined at 6 months follow-up, following completion of the intervention.
► Due to the nature of the intervention, participants will not be blinded to their treatment arm, and there is a risk that the secondary outcome self-report measures may be susceptible to reporting bias.

be disseminated through publications in research and professional journals, through conference presentations and to relevant regional and national stakeholders via online media and at dissemination events.

**Trial registration number** ISRCTN10483894.

## INTRODUCTION

Long-distance heavy goods vehicle (HGV) drivers are exposed to a multitude of health-related risk factors associated with their occupation; as a result, lorry driving has been identified as one of the most hazardous working professions.[1][2] Drivers' working environment provides limited opportunities for a healthy lifestyle and unhealthy lifestyle behaviours, such as a lack of physical activity, prolonged periods of sedentary behaviour (sitting), poor diet, a high prevalence of smoking, high volumes of alcohol

consumption and irregular sleeping patterns are widespread among this occupational group.[2–5] Furthermore, long and variable working hours, including shift work, and tight delivery schedules within the logistics and transport industry contribute to psychological stress and sleep deprivation,[6] which can lead to metabolic disturbances and further promote the uptake of unhealthy behavioural choices.[2 5–8]

Long-distance drivers exhibit higher than nationally representative rates of obesity, with observational data from a sample of HGV drivers from the UK demonstrating that 84% were overweight or obese, compared with 75% of males aged 45–54 years reported to be overweight/obese nationally.[9] Similar data have been reported from US HGV drivers.[2 10] The high rates of overweight and obesity in HGV drivers elevates their risk of numerous chronic diseases and conditions, including cardiovascular disease, type 2 diabetes, obstructive sleep apnoea and musculoskeletal disorders.[1 2 10–13]

To compound the high-risk health profile observed in long-distance drivers nationally and internationally,[1 2 10–12] the driver population in the UK (n=~300 000) has been identified as an ageing workforce (mean age: 48 years).[14] A recent parliamentary report has highlighted the 'demographic time bomb' the UK logistics industry is currently facing and the health impact of an ageing, at-risk, workforce 'driving a vehicle often referred to as 'a 40-tonne missile".[15] The UK logistics sector is also experiencing a short-fall in HGV drivers, with barriers to recruitment including the lack of roadside facilities, medical concerns and long hours of work.[16] Recommendations on how to address this shortfall and attract younger employees to the sector include increasing awareness within the industry of the need to address driver health risks and health behaviours.[15]

A systematic review[17] of health promotion interventions in truck drivers, including only eight studies, observed that the interventions generally led to improvements in health and health behaviours. However, it was concluded that the strength of the evidence was limited due to poor study designs, with no control groups, small samples and no or limited follow-up periods.[17] Since the publication of the systematic review, recent studies have examined the impacts of a weight loss intervention in US HGV drivers[18] and a smartphone application on physical activity and diet in Australian HGV drivers.[19] While positive findings were observed, the studies were limited by relatively small samples and no comparison groups.

We have developed a Structured Health Intervention For Truckers (the SHIFT programme), a multicomponent, theory-driven, health behaviour intervention designed to promote positive lifestyle changes in relation to physical activity, diet and sitting in HGV drivers. This intervention has been informed by extensive stakeholder engagement, including a qualitative study exploring the perceived barriers to healthy lifestyle behaviours in HGV drivers,[7] an observational study exploring lifestyle health-related behaviours in HGV drivers and markers of health[3]

and a pre-post pilot intervention[20] with full process evaluation.[21] Initial pre-post testing of the intervention revealed the SHIFT programme lead to favourable changes in physical activity and some markers of health.[20] This protocol paper describes a study which will build on our earlier work and generate new knowledge on the effectiveness and cost-effectiveness of the SHIFT programme, evaluated using a cluster randomised controlled trial (RCT) design with immediate and extended follow-up. Specifically, we will examine the impact of the SHIFT programme on physical activity, sedentary behaviour, fruit and vegetable intake, adiposity, sleep duration and quality, risk factors for cardiometabolic disease, psychosocial outcomes and mental health in a sample of HGV drivers.

## Study aim and objectives

The aim of this study is to evaluate the effectiveness and cost-effectiveness of the SHIFT programme using a cluster RCT.

### Primary objective

To investigate the impact of the SHIFT programme, compared with usual care, on objectively measured physical activity (expressed as steps/day) at 12 months follow-up.

### Secondary objectives

To investigate the impact of the SHIFT programme, compared with usual care, at 12 months follow-up on:
1. time spent in light and moderate-to-vigorous physical activity (MVPA);
2. sitting time;
3. measures of adiposity (body mass index (BMI), per cent body fat, waist-to-hip ratio, neck circumference);
4. blood pressure;
5. cardiometabolic risk markers (eg, glycated haemoglobin (HbA1c), total cholesterol, low-density lipoprotein-cholesterol (LDL-C), high-density lipoprotein-cholesterol (HDL-C));
6. fruit and vegetable intake;
7. sleep;
8. cognitive function and psychophysiological reactivity;
9. psychosocial variables and mental health (eg, anxiety and depression, work engagement, job performance and satisfaction, presenteeism, sickness absence, health-related quality of life and driving-related safety behaviour).

We will also conduct a full process evaluation (secondary objective 10) and a full economic evaluation (secondary objective 11).

## METHODS AND ANALYSIS
### Design

The design of this study is based on guidance from the UK Medical Research Council for developing and evaluating complex interventions,[22] and this protocol paper

has been prepared following the recommendations within the Standard Protocol Items for Randomised Trials statement.[23] This is a workplace two-armed 12-month cluster RCT, which will incorporate an internal pilot, and include both economic and process evaluations. Clusters (different worksites/depots within the same company) will be randomised, following the completion of baseline measurements, to receive either the 'SHIFT programme' or usual care condition. The impact of the intervention will be assessed at 6 and 12 months after randomisation. Figure 1 shows the overall trial design.

## Setting

This research will take place within the worksite setting of a major international logistics and transport company. The logistics and posts sector is worth approximately £55 billion to the UK economy and currently employs approximately 1.7 million people. Driving is a fundamental occupation within this industry, and drivers and warehouse workers make up the majority of the workforce within the industry.[15]

## Depot recruitment and exclusion criteria

Depots will be included in the study if they contain at least 20 long-distance HGV drivers (see 'Sample size' section). Depots containing HGV drivers who make many delivery stops, for example, drivers who deliver consumer goods to domestic customers throughout the day will be excluded. For logistical reasons, depots located within the Midlands region of the UK will be recruited. Our partner company has approximately 40 sites, containing approximately 1700 HGV drivers within this region. These sites are a similar size, and have a similar variation in size, to the company's national-level data. During recruitment, depots will be informed that they will have a 50% chance of being randomised to a current practice control condition.

## Participant recruitment and exclusion criteria

All HGV drivers within participating depots will be eligible to participate, unless they meet the following exclusion criteria: suffering from clinically diagnosed cardiovascular disease, or mobility limitations that prevent them from increasing their daily activity levels, haemophilia or have any blood-borne viruses. Posters advertising the study will be placed in participating depots for up to 4 weeks prior to the scheduling of baseline measurements. In addition, all drivers within participating depots will receive a letter and participant information sheet informing them of the study. Following the distribution of the study marketing material, researchers will visit participating depots for 1–2 days to enable interested drivers to ask any questions about the study before signing up. On completion of these visits, the researchers will provide a list of drivers' names who have agreed to participate to their Transport Managers who will then schedule time for participating drivers to attend the baseline (and follow-up) measurements.

Within the UK logistics industry, 1% of HGV drivers are women,[15] and the proportion of female HGV drivers employed by our partner company reflects this national average. While females will be included in the study, due to the small proportion of the workforce they represent, the included sample of females may not enable statistically meaningful comparisons to examine any influences of sex on the intervention. However, the sample recruited will likely reflect the gender disparities seen in the logistics and transport industry nationally and internationally.

## Sample size

Our earlier exploratory pre-post study revealed that on average HGV drivers achieve 8786 steps/day across both workdays and non-workdays with a SD of 2919 steps.[20] We have powered this study to look for a difference in step counts (the primary outcome) of 1500 steps/day (equivalent to approximately 15 min of moderately paced walking) between the intervention group and control group. Evidence demonstrates a linear association between step counts and a range of morbidity and mortality outcomes, as well as with markers of health status including inflammation and adiposity, insulin sensitivity and HDL-C in adults.[24–26] The linear association between step counts and health outcomes indicate that regardless of an individual's baseline value, even modest increases in daily step counts can yield clinically meaningful health benefits. For example, a difference in daily steps of 1500 steps/day has been associated with around a 5%–10% lower risk of all-cause mortality and cardiovascular morbidity and mortality in the general population and in those with a high risk of type two diabetes, respectively.[27 28] The proposed level of change has been chosen based on findings from our exploratory pre-post intervention,[20] while also being clinically meaningful.

Based on a cluster size of 10, a conservative ICC of 0.05 (as there is no previous data to inform this, we have been informed by recommendations of Campbell *et al*[29]), an alpha of 0.05, power of 80% and a coefficient of variation to allow for variation in cluster size of 0.51 (based on partner company data), we will require 110 participants from 11 clusters per arm. From experience in conducting such studies, it is estimated that retention and compliance rates will be approximately 70% at 12 months follow-up; therefore, the sample size will be inflated by 30% to ensure we have adequate power in our final analysis. We will also inflate the number of clusters by two to allow for whole cluster drop out. We will therefore recruit 24 clusters with an average of 14 participants per cluster.

## Intervention—the SHIFT programme

The SHIFT programme is a multicomponent lifestyle-behaviour intervention designed to target behaviour changes in physical activity, diet and sitting in HGV drivers. This 6-month intervention, grounded within the Social Cognitive Theory for behaviour change[30] consists of a group-based (four to six participants) 6-hour structured education session tailored for HGV drivers,

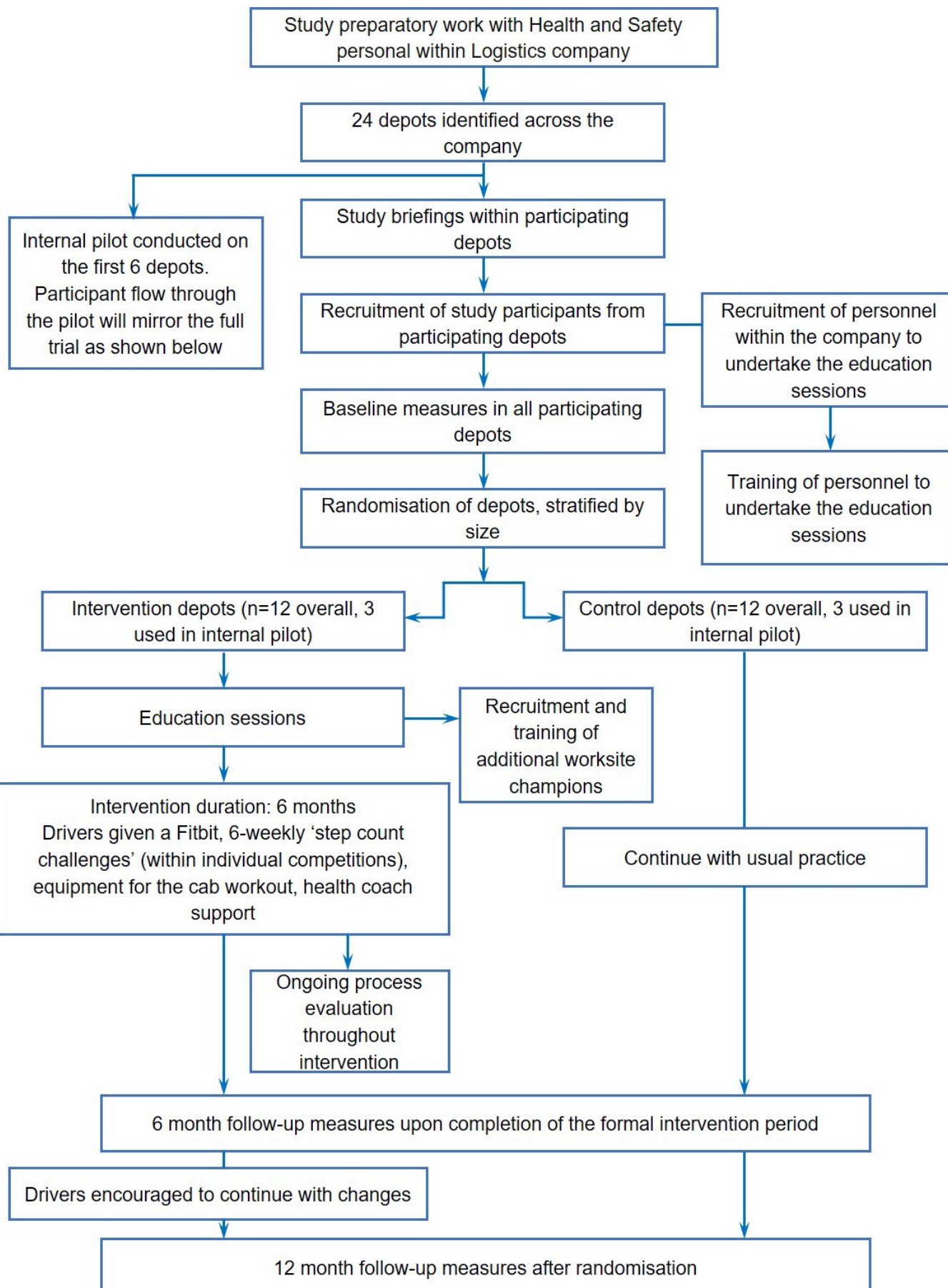

**Figure 1** Trial design and participant flow through the study.

**Table 1** Outline of the educational component of the SHIFT programme

| Section name | Theoretical underpinning | Main aims and educator activities | Duration (min) |
|---|---|---|---|
| Welcome and introduction | | Participants introduced to the SHIFT programme and made aware of both content and style of the session. | 10 |
| Driver story | Dual process theory,[74] common sense model[75] | Participants asked about their beliefs about how being a HGV driver can affect health, the causes of these health problems and controllability of these. | 30 |
| Risks and health problems | Dual process theory,[74] common sense model,[75] social learning theory[76] | Facilitator uses participant stories to support them to work out why they may be at risk of future health problems, and what to do to reduce/manage risk. | 55 |
| Physical activity | Dual process theory,[74] social learning theory[76] | Facilitator supports participants to develop knowledge and skills to support confidence to increase personal activity levels, to set personal goals and self-monitor through the use of Fitbits. Introduction and practical demonstration of the 'cab-workout'. | 80 |
| Depression, sleeping, smoking | Dual process theory,[74] social learning theory[76] | Facilitator supports participants to develop strategies to manage depression, poor sleep and smoking. | 30 |
| Food choices | Dual process theory,[74] social learning theory[76] | Facilitator supports participants to develop knowledge and skills for food choices to reduce cardiovascular risk factors and improve overall health. | 90 |
| Self-management plan | Dual process theory,[74] social learning theory[76] | Participants supported in developing personal self-management plans. | 15 |
| Questions | Common sense model,[75] social learning theory[76] | Facilitator checks all questions raised by participants throughout the programme have been answered and understood. | 5 |
| What happens next | Social learning theory[76] | Follow-up care outlined. | 5 |

HGV, heavy goods vehicle; SHIFT, Structured Health Intervention For Truckers.

delivered by two trained educators. It includes information about physical activity, diet and sitting and risk factors for type 2 diabetes and cardiovascular disease. The educational component is founded on the approach used in the award winning suite of Diabetes Education and Self-Management for Ongoing and Newly Diagnosed programme (DESMOND) programmes, including the Prediabetes Risk Education and Physical Activity Recommendation and Encouragement programme (PREPARE)[31] and Let's Prevent Diabetes programmes,[32] created by researchers at the Leicester Diabetes Centre and used throughout the National Health Service (NHS),[33] while being tailored to meet the needs of HGV drivers.[7] Within the education session participants will not be 'taught' in a formal way, but supported to work out knowledge through group discussions and to develop individual goals and plans, based on detailed individual feedback received during their health assessments (see 'Measurements' section) to achieve over the 6-month intervention period. The education session is supported by specially developed resources for HGV drivers and participant support materials. The session will include the discussion of feasible strategies for participants to increase their physical activity, improve their diet and reduce their sitting time (when not driving) during working and non-working hours. The content of the educational session is summarised in table 1.

During the education session, participants will be provided with a Fitbit Charge 2 activity tracker and encouraged to use this to set goals (agreed at the session) to gradually increase their physical activity predominately through walking-based activity. The Fitbit activity tracker will provide participants with information on their daily step counts and will be used as a tool for self-monitoring and self-regulation. Physical activity tracking using step counters (traditionally pedometers) has been associated with significant reductions in BMI and blood pressure, with interventions incorporating goal setting being the most effective.[34]

The education session will adopt the promotion of the 'small changes' philosophy using the Specific, Measurable, Attainable, Relevant, and Timely (SMART) principle[35] to encourage participants to gradually build-up their daily activity levels, within the confines of their occupation, to meet the current UK Physical Activity guidelines.[36] For example, participants will be encouraged to establish their own personalised action plan, which may also include making dietary improvements in addition to increases in physical activity, with SMART goals throughout the 6-month intervention. 'Step count challenges' (1 week competitions within intervention depots) will run every 6 weeks throughout the 6-month intervention which will be facilitated by local worksite champions. A 'cab workout' will be introduced and practised at the education session and participants will be provided with resistance bands and balls, and grip strength dynamometers to take away. Participants will be encouraged to undertake the cab workout during breaks when not permitted to leave their

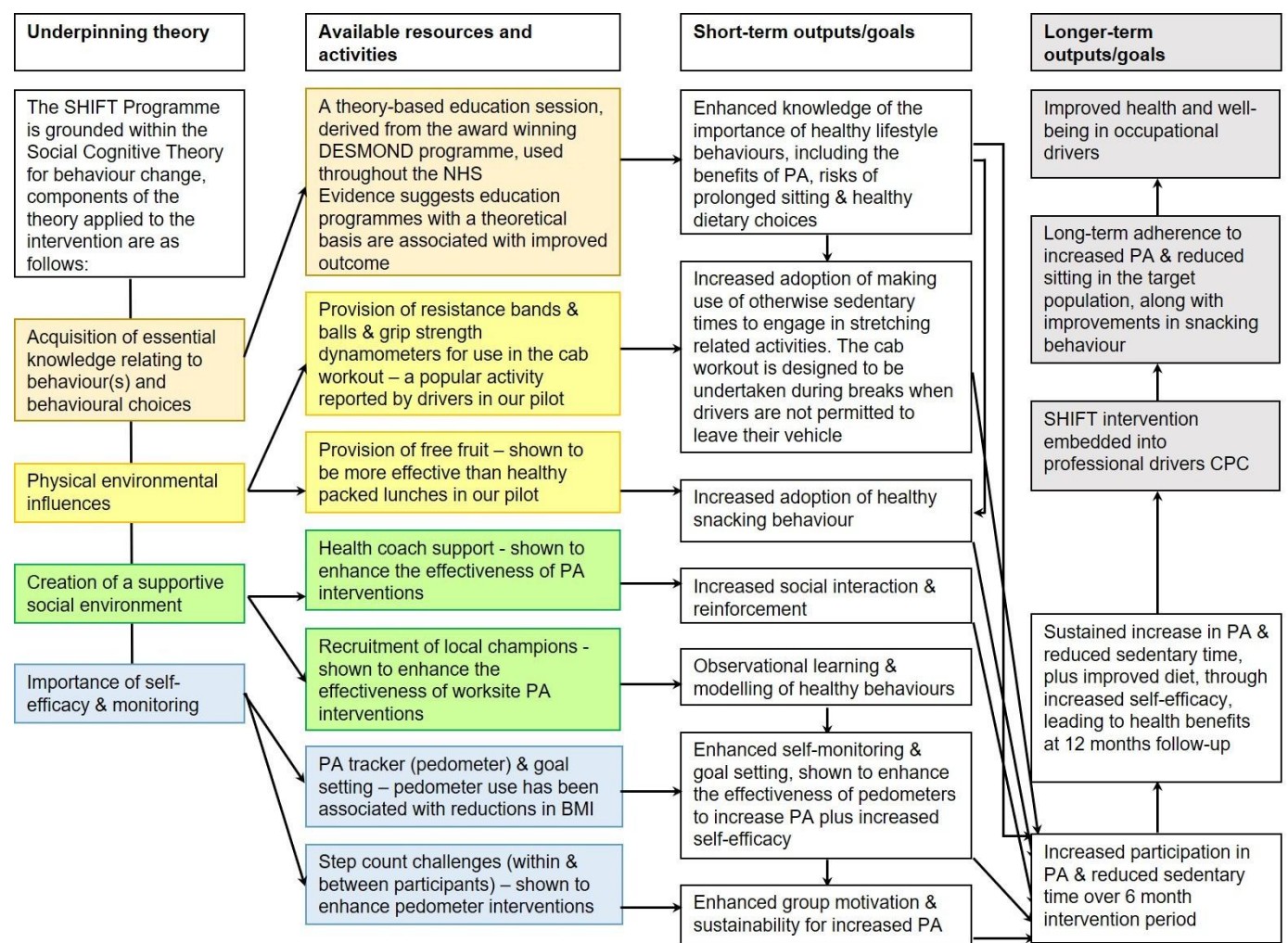

**Figure 2** Logic model for the Structured Health Intervention For Truckers (SHIFT) intervention. BMI, body mass index; CPC, continued professional competence; PA, physical activity.

vehicle. Participants will be able to keep the intervention tools beyond the 6-month intervention period, however the company will choose whether to sustain the worksite champion support and step count challenges beyond the 6-month intervention period. A logic model detailing the underlying theory behind the intervention components is shown in figure 2.

The structured education session will be delivered by trained personnel from our partner company and by trained members of the research team. These individuals will be trained and mentored by trainers from the Leicester Diabetes Centre. The education sessions will take place within appropriate training rooms within the intervention depots. Personnel delivering the education sessions within each intervention depot will also be trained to act as a local champion, shown to enhance the effectiveness of worksite physical activity interventions.[37] They will provide ongoing health coach support to intervention participants (during the 6-month intervention period) and be responsible for facilitating the step count challenges.

**Control arm**

Depots assigned to the usual practice control arm will be asked to continue with their usual care conditions.

Participants in the control depots will receive an educational leaflet at the outset detailing the importance of healthy lifestyle behaviours (ie, undertaking regular physical activity, breaking up periods of prolonged sitting and consuming a healthy diet) for the promotion of health and well-being. Control participants will be requested to complete the same study measurements as those in the intervention worksites, at the same time points. On completion of the study, control depots will be provided with all of the educational material provided to the intervention participants as part of the SHIFT programme. As the intervention will be delivered by trained personnel within our partner company, the company may choose to provide the full intervention (including the education session and health coach support) to control depots on completion of the formal trial.

**Allocation to treatment groups**

Clusters (depots within the same company) will be randomised at the worksite level into the two study arms (intervention and control, using an allocation ratio of 1:1). Randomisation into the study arms will take place in two batches; initially the first six clusters (depots) involved in the internal pilot (see below) will be randomised, and

in the second batch all of the remaining clusters will be randomised stratified by depot size. Depots will be classified as either small or large based on the median cluster size of all participating depots. In both batches randomisation will take place on completion of baseline measurements and will be done by an independent statistician at the Leicester Clinical Trials Unit.

## Measurements

The outcome measurements will be assessed at three time points. Baseline measures will occur prior to randomisation of the depots into the two study arms. A second set of identical measurements will take place 6 months postrandomisation (ie, just after the completion of the 6-month intervention), and a final set will be taken at 12 months postrandomisation to assess the sustainability of the intervention (ie, 6 months after completion of the formal intervention period, as recommended by the National Obesity Observatory).[38] At the baseline assessment, the study will be explained to the participant and written informed consent will be obtained. The measurements will be undertaken in suitable rooms within participating depots by trained researchers and will last between 1.5 and 2 hours per participant. Participants will complete a range of self-report questionnaires and have a series of physiological health assessments taken. All participants will receive detailed feedback on their physiological health assessment measures during each measurement session. In the event that a potential health issue is evident during the health assessments, such as undiagnosed hypertension or high cholesterol levels, participants will be advised to visit their GP for further checks. We will provide participants with a letter to give to their GP which summarises the findings from our point-of-care (blood markers) and automated (blood pressure) measures. Participants will be requested to inform the researchers about the use of any prescribed medications that they commence throughout the study duration which may impact the proposed outcome measures. Participants will be issued with objective monitoring devices to assess their free-living physical activity, sedentary behaviour and sleep, which they will be instructed to wear for 8 days following each measurement visit. After 8 days, participants will be requested to return these monitors to their depot where they will be collected by a member of the research team.

### Primary outcome

The primary outcome will be physical activity, expressed as steps/day, at 12 months postrandomisation. Physical activity will be objectively measured using the activPAL microaccelerometer, worn continuously on the anterior aspect of the thigh, for 24 hours/day over 8 days during each assessment period. The activPAL provides a valid measure of walking and posture (ie, sitting and standing) in adults,[39–41] and provides a more accurate measure of physical activity and sitting in occupational drivers in comparison to waist-worn accelerometers.[42] As the physical activity component of the intervention predominantly includes the promotion of walking-based activity, and as participants will be provided with a Fitbit providing information on daily step counts to set goals to increase their physical activity, steps/day was chosen as the primary physical activity-related outcome.

### Secondary outcomes

A number of secondary outcomes will be assessed at all measurement time points. The secondary outcomes are described as follows:

#### Physical activity and sedentary behaviour

Light physical activity and MVPA will be assessed using the activPAL and the wrist-worn GENEActiv accelerometer, both worn continuously for 8 days. The GENEActiv is a lightweight waterproof device, resembling a sports watch, which has been found to be a valid and reliable objective measure of physical activity.[43] Outcomes calculated from the GENEActiv include minutes spent in MVPA, proportion of participants meeting the MVPA guidelines of 150 min/week, total volume of physical activity regardless of intensity, and sleep duration. The accelerometer provides time-stamped data so activity at specific times of the day (eg, during work, after work) will also be extracted.

Sedentary behaviour will also be measured for eight consecutive days during each assessment period using the activPAL3 micro. The activPAL is regarded as the most accurate method of assessing sitting behaviour in free-living settings,[41 44 45] and is recommended for use in interventions when sitting is an outcome measure.[40] From the data provided, we will extract total daily sitting time, work-time and leisure-time sitting, sitting bout durations and number of transitions between sitting and standing.

#### Sleep duration, subjective sleepiness and chronotype

Sleep duration and efficiency will be measured objectively using the GENEActiv which has been shown to be an accurate measure of sleep, in addition to physical activity.[46] Subjective sleepiness will be assessed using the Karolinska Sleepiness Scale, shown to be a valid measure of sleepiness when validated against electroencephalography and performance outcomes.[47 48] Participants' chronotype will be determined using the short version of the Morningness-Eveningness Questionnaire.[49]

#### Anthropometry, adiposity and blood pressure

Stature (measured at baseline only) and body mass (both assessed without shoes), along with waist and hip circumferences, will be measured using standardised anthropometric techniques by trained research staff. BMI will be calculated as weight (kg)/height (m$^2$). Body composition (percentage body fat and fat mass) will be assessed via bioimpedance analysis, using Tanita DC-360S body composition scales. We will also measure neck circumference which is a novel marker which links strongly to obstructive sleep apnoea, insulin resistance and cardiovascular disease risk.[50] Blood pressure will be measured from the left arm after a 20 min period of quiet sitting

using an automated recorder (Omron HEM-907), in accordance with current recommendations.[51]

### Biochemical assessments

Finger-prick blood samples will be collected from participants, with participants being requested to fast for ≥4 hours prior to attending each health assessment. The 'A1CNow+' point-of-care analyser will be used to measure HbA1c which is a marker of long-term glucose regulation used in clinical care. Additionally, we will use the Cardiochek point-of-care analyser to measure circulating cholesterol (total, HDL, LDL). Both of these systems are manufactured by PTS Diagnostics and possess analyte validation certificates from the International Federation of Clinical Chemistry and Laboratory Medicine.

### Functional fitness

Grip strength will be assessed from both hands using the Takei Hand-Grip dynamometer (Takei Scientific Instruments, Japan). Reduced muscular strength, as measured by grip strength, is associated with an increased risk of cardiovascular disease, and all-cause and cardiovascular mortality.[52]

### Cognitive function and psychophysiological reactivity

The Stroop test will be administered over a 5 min period using a validated software package to provide a measure of reaction time, sensitivity to interference and the ability to suppress an automated response—reading colour names in favour of naming the font colour.[53] To examine psychophysiological reactivity, acute stress will be induced using a 5 min mirror-tracing task (Campden Instruments), during which measures of blood pressure and heart rate will be taken.[54]

### Work-related psychosocial variables and mental health

A series of self-report measures will be employed to characterise work-related health and mental health: musculoskeletal symptoms will be assessed using the Standardised Nordic Questionnaire[55]; work engagement (characterised by vigour, dedication and absorption) will be measured using the Utrecht Work Engagement Scale[56]; occupational fatigue will be measured using the Occupational Fatigue Exhaustion Recovery 15 scale[57]; job performance[58] and job satisfaction[59] will be measured using single-item 7-point Likert scales; sickness presenteeism will be assessed using a single-item questionnaire; participant's perceptions of work demand and support will be assessed using four subscales from the Health and Safety Executive Management Standards Indicator Tool,[60] and driving-related safety behaviour will be assessed using a 6-item measure.[61] Anxiety and depression will be measured using the Hospital Anxiety and Depression Scale,[62] and social isolation will be assessed using the 8-item Social Isolation short form from the Patient-Reported Outcomes Measurement Information System.[63 64] Data on sickness absence will be collected via self-report and employer records and will include frequency and duration of self-certified and certified

sickness. Data on sickness absence will be collected from organisational records for 12 months prior to the intervention and for the 6-month intervention and follow-up periods.

### Health-related quality of life and health-related resource use

The self-reported EQ5D[65] will be completed by participants during each assessment period to inform the within-trial cost-effectiveness analysis (see 'Cost-effectiveness' section). Participants will also complete a questionnaire, developed for this study, assessing health-related resource use at the same time points.

### Demographics and additional lifestyle health-related behaviour measures

At baseline we will collect basic demographic information for each participant including their date of birth, sex, ethnicity, highest level of education, marital status, postcode (to determine Index of Multiple Deprivation as an indicator of neighbourhood socioeconomic status), working hours, years worked as a HGV driver and years worked at our partner company. At each follow-up assessment, participants will be asked if there have been any changes in these variables. During each assessment, information on smoking status and typical alcohol intake will be gathered by self-report measures. Dietary quality, including fruit and vegetable intake, will be assessed using a short-form food frequency questionnaire.[66]

### Internal pilot

We intend to conduct an internal pilot study using the first six clusters (depots). The internal pilot will examine issues surrounding worksite and participant recruitment, randomisation, compliance to the primary outcome and retention rates at 6 months following randomisation. After this period, we will continue to the full trial if the following progression criteria are met:

► All 24 depots required for the full sample size agree to take part in the study. Six depots will be selected to take part in the internal pilot (three will be randomised to the intervention arm and three to the control arm). This will demonstrate that depot recruitment and intervention delivery is on track.
► According to our criteria, 84 drivers will need to agree to participate in the internal pilot, based on an average of 14 participants per cluster.
► An average of 75% of drivers opting into the study, randomised into the intervention arm, attend the education session across the three intervention depots. This figure is based on the intervention uptake rate seen in our exploratory pre-post intervention study (87%),[20] while also recognising that take-up rates tend to be lower when moving from an efficacy to a larger multicentre effectiveness trial.
► No more than 20% of participants fail to provide valid data for the primary outcome measure (activPAL-determined step counts) at baseline and at 6 months postrandomisation or withdraw/are lost to follow-up

during the 6-month intervention phase. This threshold is necessary as study power requires total withdrawal or loss to follow-up of no higher than 30% during the 6-month intervention and 6-month follow-up (12 months postrandomisation).

If the final two progression criteria are not fully met, strategies to improve these metrics for the full trial will be discussed with the Trial Steering Committee and the study will progress based on recommendations from this committee.

## Process evaluation

The process evaluation will be used to help explain any discrepancies between expected and observed outcomes, to understand the influence of intervention components and context on the observed outcomes and to provide insight for any further intervention development and implementation.[22] Throughout the intervention, we will monitor the reach, efficacy, adoption, implementation and maintenance of the intervention using the reach, efficacy, adoption, implementation, and maintenance (RE-AIM) framework.[67] We will employ a variety of techniques (eg, logbooks, questionnaires, interviews and focus groups) to inform our process evaluation. For example, Transport Managers (or their nominated facilitators) and educators/worksite champions from each site will report on a monthly basis if there were any organisational changes (eg, job changes) or events which may affect participation. Self-report questionnaires provided to study participants will evaluate the various intervention components (eg, education session, physical activity monitoring tool, cab workout). Interviews and focus groups with study participants will further examine engagement in the various components of the intervention, along with any perceived barriers or facilitators to participating in these components. Interviews and focus groups with worksite champions, HR staff, health and safety personnel and logistics timetabling and planning staff will further examine the intervention implementation. We will also document any environmental factors (eg, movement of personnel between worksites/depots, potential contamination of the intervention through drivers in different groups meeting at service stations/customer distribution centres) that may have an influence on intervention effectiveness. Details of the process evaluation components are included in table 2.

## Patient and public involvement

This trial is the result of an earlier 3-year partnership between the research team and a large transport and logistics company (different to our partner in the present study) in the East Midlands, UK. The preparatory work which informed this study[3 7 16 17] was instigated by the company who requested help in improving the lifestyle behaviours and health of their long-distance drivers who were proving difficult to engage. The SHIFT programme was developed in collaboration with long-distance HGV drivers and health and safety personnel working within the logistics sector. Following pilot testing, the intervention and outcome measures described within this protocol have been refined following further input from drivers and associated stakeholders. A driver and manager working within the logistics sector will sit on our independent Trial Steering Committee and will provide invaluable insight into the design, set-up, conduct and dissemination of this research. Throughout the trial, we will conduct regular patient and public involvement events with relevant stakeholders to gain feedback on the trial's progress. The research team will also continue to work with the Chartered Institute of Logistics and Transport to facilitate research dissemination (articles, conferences, workshops) across the logistics and transport sector nationally and internationally.

## Data analysis

### Statistical analysis: internal pilot

The average recruitment rate across depots, proportion of participants providing valid data and attendance rate at the education sessions will be reported with 95% CI. The point estimates and 95% CIs will be compared with the progression criteria outlined outlined in the Internal Pilot section

### Statistical analysis: main trial

Average daily steps at 12 months will be compared by group using generalised estimating equation models adjusted for baseline values and waking wear time with an exchangeable correlation structure, which adjusts for clustering. For the primary analysis missing data will not be replaced (complete case analysis) but participants will be included in the intervention group in which their depots were randomised irrespective of the intervention actually received (modified intention-to-treat analysis). We have inflated our sample size by 30% to account for potential loss to follow-up and non-compliance with the primary outcome measure. We will compare the baseline characteristics of those who have complete primary outcome data and those who do not. A sensitivity analysis using multiple imputation will be performed to assess the impact of missing outcome data on the results found and to account for uncertainty associated with imputing data (full intention-to-treat analysis). The imputation will be carried out using the command MI in Stata. MI replaces missing values with multiple sets of simulated values to complete the data, performs standard analysis on each completed dataset and adjusts the obtained parameter estimates for missing data uncertainty using Rubin's rules to combine estimates. The effect size will also be assessed by attendance excluding those who did not attend the full intervention (per-protocol analysis). Secondary outcomes and 6month data will be analysed using similar methodology.

### Qualitative analyses

Audio-recordings of interviews and focus groups with drivers, worksite champions, HR staff, health and safety personnel and logistics timetabling and planning staff will be transcribed verbatim and analysed using framework

**Table 2** Process evaluation plan for the SHIFT intervention

| Areas to measure | General process questions | Data source and data collection method | Total numbers and sampling strategy/ timescales |
|---|---|---|---|
| Recruitment | Number of depots/worksites invited to participate, and number agreeing. Number of possible participants at each depot, number invited/recommended for participation, number opting in to the intervention. Number of participants opting-out, dropping out and non-compliance to the primary outcome measure. | Project records, including the number of drivers within each depot approached. Depot logs of staff numbers, project records, attendance records at measurements. Participant attendance records, short questionnaires to explore reasons for non-participation, dropping out and non-compliance. | Ongoing throughout the project. |
| Acceptability of randomisation and measurement tools | How depots feel about being randomised to intervention/control arms? Did participants find outcome assessments acceptable? How did participants and logistics timetabling staff experience recruitment and timetabling of outcome assessments? | Focus groups with participants. Interviews with local depot health and safety advisors/HR/timetabling staff. | ~8 focus groups, or until data saturation is reached, with participants ~1 month following completion of baseline measures. ~8 interviews, or until data saturation is reached, with local depot health and safety advisors/HR/timetabling staff ~1 month after completion of baseline measures in their depots. |
| Intervention acceptability and fidelity—implementation | Was the intervention implemented as planned? How did participants and logistics timetabling staff experience scheduling the education sessions? | Interviews with personnel within our logistics partners who are trained as educators and implemented the education sessions. Interviews with local worksite champions and timetabling staff within intervention depots. Participant questionnaires. | Interviews with educators, the number of which will depend on the number of educators trained, and timetabling staff immediately following delivery of the education sessions. Interviews with local champions 3 months into the intervention, immediately following the intervention (6 months), and at 9 and 12 months. Questionnaires administered after education sessions to participants. |
| Intervention acceptability and fidelity—participation | What proportion of the target group participated in the intervention, and what components of the intervention were preferred, did this differ between males and females? What strategies were put in place by intervention participants to facilitate behaviour change? | Focus groups with intervention participants. Attendance logs at education sessions and measurement visits. Questionnaires and focus groups. | ~8 focus groups, or until data saturation is reached, with participants immediately following completion of the intervention (6 months). Brief questionnaires administered to all intervention participants at 6 months during health assessments. |
| Intervention sustainability | What proportion of the target group maintained any changes in their health behaviours following the 6-month intervention period? Were there any differences in sustainability between males and females? Are the company going to continue with the intervention in some way? | Focus groups with intervention participants. Questionnaires. Interviews with health and safety personnel. | ~8 focus groups, or until data saturation is reached, with participants at 10 months follow-up (4 months after completion of the intervention). Brief questionnaires administered to all intervention participants at 12 months during health assessments. Interviews at 12 months. |
| Intervention contamination | Did movement of staff (eg, participants, health and safety personnel) occur from intervention to control depots? Did intervention drivers interact with control drivers at customer warehouses/distribution centres, etc? | Control depots to keep a log of any staff changes. Focus groups with intervention and control participants. | Logs collected on completion of the 12-month follow-up assessments. 8 focus groups, or until data saturation is reached, with intervention and control participants immediately following completion of the intervention (6 months) and at 10 months follow-up. |
| Unexpected events arising from the study | Did intervention and control participants modify their behaviours based on information provided at the baseline health assessments? Did the health assessments prompt general practitioner visits? Did increased self-awareness of health status and constraints within the job lead to cognitive dissonance? Did intervention participants change an existing activity-related behaviour for another as a result of participating in the study? | Focus groups, interviews and questionnaires delivered to intervention and control participants. | Questionnaires delivered to intervention and control participants 1 month after completion of the baseline health assessments. 8 focus groups, or until data saturation is reached, with intervention and control participants immediately following completion of the intervention (6 months) and at 10 months follow-up. One-to-one interviews based on questionnaire and focus group responses at 1 and 10 months. |

analysis,[68 69] using the RE-AIM framework[67] as the overarching framework.

## Cost-effectiveness

The economic analysis will consist of a cost-consequence analysis based on the observed results within the trial period and a cost-effectiveness analysis where differences between groups in the trial will be extrapolated to the longer term. For both analyses, costs in both arms will be estimated from a NHS and Personal Social Services perspective (consistent with that used by the National Institute for Health and Care Excellence) as well as a wider public sector perspective. In each analysis, the cost of the SHIFT arm will include an estimate of the cost of the intervention (including the cost of training the educators), generated through a staff questionnaire completed at the end of each education session.

## Within-trial analysis

Within the trial, resource use estimates will be collected from participant questionnaires and will include health-related resource use as well as absence from employment. The health-related resource use will be based on a variant of the Client Service Receipt Inventory and will include services that this population are likely to use such as GPs and practice nurse appointments, occupational health visitors and counsellors. Costs of resources will be calculated by applying published national unit cost estimates (eg, NHS reference costs or Personal Social Services Research Unit costs of health and social care),[70 71] where available, to estimates of relevant resource use.

A range of outcomes will be assessed in the trial including health-related quality of life, measured using the EQ5D.[65] The within-trial analysis will present incremental results for the primary and secondary outcomes (including EQ5D) in both intervention and control arms and will be compared with the incremental costs measured. We will also present the results in terms of the differences between the groups in time absent from work. Two analyses will be conducted, one including these productivity losses, the other excluding them. This will allow decision makers to assess the importance of inclusion of these costs in the adoption decision.

## Long-term analysis

It is acknowledged that although there may be short-term health benefits from the intervention, the long-term effects of, for example, increased physical activity on diabetic status and number of cardiovascular events may be more important. We will therefore conduct a brief literature review to identify existing models that link short-term end points (including physical activity) measured in the trial and long-term quality of life. We have identified and used existing models[72] linking physical activity to quality-adjusted life years (QALYs) previously. These models will be used to extrapolate costs and effects of the intervention beyond the trial period to a more appropriate time horizon. If appropriate an incremental cost-effectiveness ratio for the extrapolated period will be reported using the QALY. As with the within-trial analysis, we will conduct analyses where productivity losses are included/excluded to assess the impact on decision making. Costs and effects will be discounted at the prevailing recommended rate (currently 1.5% per annum on both costs and effects), but will be the subject of sensitivity analysis to reflect the ongoing uncertainty around appropriate discount rates for public health interventions. To reflect the levels of uncertainty in parameter inputs we will conduct probabilistic sensitivity analyses; this will allow a characterisation of the uncertainty around the adoption decision which we will depict using cost-effectiveness acceptability curves. Sensitivity analyses will be performed to determine the robustness of the results to altering certain assumptions such as the discount rate or inclusion/exclusion of productivity losses.

## DISCUSSION

HGV drivers' working environments are not conducive to a healthy lifestyle, despite this they are currently an underserved occupational group in terms of health promotion efforts and exhibit higher than nationally representative rates of obesity and related comorbidities.[10 17] The health and well-being of professional drivers is of public concern given their health impacts the safety of all road users.[5 17] Of concern, obese HGV drivers are approximately 50% more likely to have an accident than normal weight drivers,[73] with accident rates increasing further in HGV drivers suffering from obstructive sleep apnoea, a prevalent condition in this occupational group.[8] The present study will target health-related behaviours of this at-risk and underserved occupational group, with the goal of making a positive long-term impact on long-distance HGV drivers' health. Given the current absence of resources available to tackle health inequalities within the transport sector, it is anticipated that if effective, the SHIFT programme could be scalable as a Continued Professional Competence resource for HGV drivers nationally and internationally. This resource will likely be modifiable for use across other driving-related occupations. This could have a long-term impact on professional drivers' health, lead to cost savings within the logistics and transport sector and ultimately impact road safety for all road users.

To our knowledge, this is the first RCT to examine the impact of a multicomponent intervention targeting individual and environmental barriers faced by HGV drivers to lead a healthy lifestyle. Strengths of the study include the robust RCT design, with randomisation at the depot/cluster level to reduce contamination. The large fully powered sample, objectively derived outcome measures and 6month follow-up assessment after completion of the SHIFT programme are further strengths, along with the extensive process and economic evaluations.

**Author affiliations**
[1]School of Sport, Exercise and Health Sciences, Loughborough University, Loughborough, UK
[2]Diabetes Research Centre, University of Leicester, Leicester, UK
[3]Department of Health Sciences, University of Leicester, Leicester, UK
[4]Leicester Clinical Trials Unit, University of Leicester, Leicester, UK
[5]Centre for Health Economics, University of York, York, UK
[6]Leicester Diabetes Centre, University Hospitals of Leicester NHS Trust, Leicester, UK

**Contributors** All authors have made substantial contributions to the concept and design of the study. The study was conceived by SAC, VVM, JAK, TY, CLE, MH, FM, LJG, JT, VJ and GR. VVM, VJ and Y-LC are the qualified Research Associates on the project. LJG and NBJ are the trial statisticians and developed the statistical analysis plan and GR developed the cost-effectiveness plan. The first draft of this manuscript was written by SAC with input from all other authors. All authors have edited and critically reviewed the paper for intellectual content and approved the final version of the paper.

**Funding** This study is funded by the NIHR Public Health Research Programme (reference: NIHR PHR 15/190/42) and supported by the NIHR Leicester Biomedical Research Centre – Lifestyle theme. Funding to cover intervention costs (Fitbits, cab workout equipment) has been received from the Higher Education Innovation Fund via the Loughborough University Enterprise Projects Group.

**Competing interests** None declared.

**Patient consent for publication** Not required.

**Ethics approval** Ethical approval was obtained from the Loughborough University Ethics Approvals (Human Participants) Sub-Committee (reference: R17-P063). Study findings will be disseminated through publications in research and professional journals, through conference presentations at scientific meetings, and to relevant regional and national stakeholders via online media and at dissemination events. Loughborough University are the trial sponsor.

**Provenance and peer review** Not commissioned; externally peer reviewed.

**ORCID iD**
Stacy A Clemes http://orcid.org/0000-0001-5612-5898

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
