## [Reviewer comments · BMJ Open]

ARTICLE DETAILS

TITLE (PROVISIONAL)	A cluster randomised controlled trial to investigate the effectiveness and cost-effectiveness of a Structured Health Intervention For Truckers (The SHIFT Study): Study Protocol
AUTHORS	Clemes, Stacy; Varela Mato, Verónica; Munir, Fehmidah; Edwardson, Charlotte; Chen, Yu-Ling; Hamer, Mark; Gray, Laura; Bhupendra Jaicim, Nishal; Richardson, Gerry; Johnson, Vicki; Troughton, Jacqui; Yates, Thomas; King, James

VERSION 1 – REVIEW

REVIEWER	Sergio Garbarino Neuroscienze Department, University of Genoa Italy
REVIEW RETURNED	05-Apr-2019

GENERAL COMMENTS	The authors deal with one of the most important topics for public health and safety at a global level. The working environments are not conducive to a healthy lifestyle, yet there has been limited attention to health promotion efforts. In this paper they describe the protocol of a cluster randomised controlled trial designed to evaluate the effectiveness and cost-effectiveness. The authors have developed a Structured Health Intervention For Truckers (the SHIFT programme), a multicomponent, theory-driven, health-behaviour intervention targeting physical activity, diet, and sitting in heavy goods vehicle drivers. Authors should complete the reference bibliography such as in the case of Sleep duration, subjective sleepiness and chronotype and in the discussion as for example: 1: Garbarino S, Guglielmi O, Sannita WG, Magnavita N, Lanteri P. Sleep and Mental Health in Truck Drivers: Descriptive Review of the Current Evidence and Proposal of Strategies for Primary Prevention. Int J Environ Res Public Health. 2018 Aug 27;15(9). pii: E1852. doi: 10.3390/ijerph15091852. 2: Garbarino S, Magnavita N, Guglielmi O, Maestri M, Dini G, Bersi FM, Toletone A, Chiorri C, Durando P. Insomnia is associated with road accidents. Further evidence from a study on truck drivers. PLoS One. 2017 Oct 31;12(10):e0187256. doi: 10.1371/journal.pone.0187256. eCollection 2017. 3: Garbarino S, Durando P, Guglielmi O, Dini G, Bersi F, Fornarino S, Toletone A, Chiorri C, Magnavita N. Sleep Apnea, Sleep Debt and Daytime Sleepiness Are
---

	Independently Associated with Road Accidents. A Cross-Sectional Study on Truck Drivers. PLoS One. 2016 Nov 30;11(11):e0166262. doi:10.1371/journal.pone.0166262. eCollection 2016. 4: Garbarino S, Guglielmi O, Campus C, Mascialino B, Pizzorni D, Nobili L, Mancardi GL, Ferini-Strambi L. Screening, diagnosis, and management of obstructive sleep apnea in dangerous-goods truck drivers: to be aware or not? Sleep Med. 2016 Sep;25:98-104. doi: 10.1016/j.sleep.2016.05.015
--	--

REVIEWER	Sergio Useche University of Valencia, Spain
REVIEW RETURNED	08-Apr-2019

GENERAL COMMENTS	After a careful review of the submitted protocol, I found that most of the basic aspects related to form, structure and contents are adequately presented. Moreover, the topic addressed by the paper is worth of investigation. Thus, I should say that the manuscript has not only potential, but pertinence for being considered to be published in BMJ Open. However, some few queries raised along my review (most of them relatively minor and/or amendable. Please check below:  1. At the introduction, you summarized some circumstances and factors that potentially impair both the health and performance of professional drivers. This set of factors include key variables such as stress, fatigue and shift work. However, the supporting evidence in this regard is notoriously scarce, and a major effort should be done by the authors to attach evidences on how these processes negatively affect the performance and well-being of this vulnerable occupational group. And this is a crucial task to be performed, in order to increase the pertinence and validity of the developed protocol. For four guidance, please check empirical sources such as the suggested here: (DOIs): 10.1371/journal.pone.0207322, 10.7717/peerj.6249, 10.1016/j.aap.2017.04.023 2. As authors also included the use of subjective and self-report-based measures (e.g., KSS), the potential confounding role of CMBs and CMV should be stated at the limitations. 3. Another critical aspect that the protocol may address is the worldwide prevalent state of affairs in gender disparities in professional transportation. Although high ratios of finding excessively big proportions of male workers in the transport industry does not invalidate the potentially collected data, it might bias the interpretation on it (considering factors such as differences in psychophysiological parameters and psychosocial variables). Therefore, my suggestion is to include (apart that is already stated for intervention phase) a paragraph/sub-section stating this potential scenario, and some directions for managing gender-based data. Some of the suggested sources also mention this fact. 4. Qualitative analysis (pages 22-23) need a bit more of clearness and systematicity to be presented. Guidelines in this regard are scarce and may difficult the further appliance of the protocol.
---

REVIEWER	Inna Feldman Uppsala University, Sweden
-----------------	--

REVIEW RETURNED	11-May-2019
-------------

GENERAL COMMENTS	This is a very interesting study describing the planned evaluation of a health-behaviour promotion intervention targeting physical activity, diet, and sitting in heavy goods vehicle drivers. My focus was a part concerning cost-effectiveness analysis and I found some areas of the manuscript that would benefit from revision/clarifications as noted below Longer-term analysis (page 25) This is very correct that we can expect the longer-term effects of the intervention that could be more important from cost-effectiveness point of view than short-term but it is not completely clear how the researcher are going to estimate that. According to the manuscript, they will “conduct a brief literature review to identify existing models that link short term endpoints measured in the trial and longer-term quality of life”. Questions:  1) Which endpoints measures are they going to use? 2) Is that enough to link those measures to QALY – you probably need the related long-term costs? 3) Can you get an access to the appropriate model through the literature Review, to imput the necessary parameters obtained from the trial? I suggest clarifying the modelling aspect in the long-term analysis
---

VERSION 1 – AUTHOR RESPONSE

Reviewer: 1

The authors deal with one of the most important topics for public health and safety at a global level. The working environments are not conducive to a healthy lifestyle, yet there has been limited attention to health promotion efforts. In this paper they describes the protocol of a cluster randomised controlled trial designed to evaluate the effectiveness and cost-effectiveness. The authors have developed a Structured Health Intervention For Truckers (the SHIFT programme), a multicomponent, theory-driven, health-behaviour intervention targeting physical activity, diet, and sitting in heavy goods vehicle drivers. Authors should complete the reference bibliography such as in the case of Sleep duration, subjective sleepiness and chronotype and in the discussion as for example:

1: Garbarino S, Guglielmi O, Sannita WG, Magnavita N, Lanteri P. Sleep and Mental Health in Truck Drivers: Descriptive Review of the Current Evidence and Proposal of Strategies for Primary Prevention. *Int J Environ Res Public Health*. 2018 Aug 27;15(9). pii: E1852. doi: 10.3390/ijerph15091852.

2: Garbarino S, Magnavita N, Guglielmi O, Maestri M, Dini G, Bersi FM, Toletone A, Chiorri C, Durando P. Insomnia is associated with road accidents. Further evidence from a study on truck drivers. *PLoS One*. 2017 Oct 31;12(10):e0187256. doi: 10.1371/journal.pone.0187256. eCollection 2017.

3: Garbarino S, Durando P, Guglielmi O, Dini G, Bersi F, Fornarino S, Toletone A, Chiorri C, Magnavita N. Sleep Apnea, Sleep Debt and Daytime Sleepiness Are Independently Associated with Road Accidents. A Cross-Sectional Study on Truck Drivers. *PLoS One*. 2016 Nov 30;11(11):e0166262. doi:10.1371/journal.pone.0166262. eCollection 2016.

4: Garbarino S, Guglielmi O, Campus C, Mascialino B, Pizzorni D, Nobili L, Mancardi GL, Ferini-Strambi L. Screening, diagnosis, and management of obstructive sleep apnea in dangerous-goods truck drivers: to be aware or not? *Sleep Med*. 2016 Sep;25:98-104. doi: 10.1016/j.sleep.2016.05.015

We would like to thank the Reviewer for drawing our attention to these interesting and relevant publications, where appropriate we have referenced the papers linked to sleep apnoea and risk factors for poor mental health and sleep conditions in the introduction (page 4, paragraphs 1 and 2) and discussion (page 25, discussion paragraph 1). We have also added a comment in the discussion about the risk of accidents being dramatically increased in HGV drivers with obstructive sleep apnoea (page 25, discussion paragraph 1).

Reviewer: 2

After a careful review of the submitted protocol, I found that most of the basic aspects related to form, structure and contents are adequately presented. Moreover, the topic addressed by the paper is worthy of investigation. Thus, I should say that the manuscript has not only potential, but pertinence for being considered to be published in BMJ Open.

However, some few queries raised along my review (most of them relatively minor and/or amendable). Please check below:

1. At the introduction, you summarized some circumstances and factors that potentially impair both the health and performance of professional drivers. This set of factors include key variables such as stress, fatigue and shift work. However, the supporting evidence in this regard is notoriously scarce, and a major effort should be done by the authors to attach evidences on how these processes negatively affect the performance and well-being of this vulnerable occupational group. And this is a crucial task to be performed, in order to increase the pertinence and validity of the developed protocol. For guidance, please check empirical sources such as the suggested here: (DOIs): [10.1371/journal.pone.0207322](https://doi.org/10.1371/journal.pone.0207322), [10.7717/peerj.6249](https://doi.org/10.7717/peerj.6249), [10.1016/j.aap.2017.04.023](https://doi.org/10.1016/j.aap.2017.04.023)

Based on the comment above, and the comment from Reviewer 1, we have modified the first paragraph slightly and included further evidence in the form of citations to strengthen the rationale for promoting healthier lifestyle behaviours in HGV drivers.

2. As authors also included the use of subjective and self-report-based measures (e.g., KSS), the potential confounding role of CMBs and CMV should be stated at the limitations.

We have added that there may be a risk of reporting bias on the secondary outcome self-report measures in the 'Strengths and limitations of this study' section of the manuscript.

3. Another critical aspect that the protocol may address is the worldwide prevalent state of affairs in gender disparities in professional transportation. Although high ratios of finding excessively big proportions of male workers in the transport industry does not invalidate the potentially collected data, it might bias the interpretation on it (considering factors such as differences in psychophysiological parameters and psychosocial variables). Therefore, my suggestion is to include (apart that is already stated for intervention phase) a paragraph/sub-section stating this potential scenario, and some directions for managing gender-based data. Some of the suggested sources also mention this fact.

We have added the following text to page 9 to acknowledge the gender disparities seen in the Logistics and Transport industry in the UK, and internationally:

"Within the UK logistics industry, 1% of HGV drivers are women,¹⁵ and the proportion of female HGV drivers employed by our partner company reflects this national average. Whilst females will be included in the study, due to the small proportion of the workforce that they represent, the included sample of females may not enable statistically meaningful comparisons to examine any influences of sex on the intervention. However, the sample recruited, will likely reflect the gender disparities seen in the Logistics and Transport industry nationally and internationally."

4. Qualitative analysis (pages 22-23) need a bit more of clearness and systematicity to be presented. Guidelines in this regard are scarce and may difficult the further appliance of the protocol.

We have removed some small pieces of text from the process evaluation section on pages 20-21 to simplify this section and make it clearer.

Reviewer: 3

This is a very interesting study describing the planned evaluation of a health-behaviour promotion intervention targeting physical activity, diet, and sitting in heavy goods vehicle drivers.

My focus was a part concerning cost-effectiveness analysis and I found some areas of the manuscript that would benefit from revision/clarifications as noted below

Longer-term analysis (page 25)

This is very correct that we can expect the longer-term effects of the intervention that could be more important from cost-effectiveness point of view than short-term, but it is not completely clear how the researcher are going to estimate that. According to the manuscript, they will “conduct a brief literature review to identify existing models that link short term endpoints measured in the trial and longer-term quality of life”.

Questions:

1) Which endpoint measures are they going to use?

We had anticipated exploring links between several short-term outcomes and longer-term outcomes, depending on the availability of literature and models. We have previously used the MOVES model to link physical activity to QALYs and that is our starting point

2) Is that enough to link those measures to QALY – you probably need the related long-term costs?

Thank you for pointing this out. We will be considering costs and QALYs and have amended the text on page 24 accordingly, this text now reads:

“We will therefore conduct a brief literature review to identify existing models that link short term endpoints (including physical activity) measured in the trial and longer-term quality of life. We have identified and utilised existing models⁷² linking physical activity to Quality Adjusted Life Years (QALYs) previously. These models will be utilised to extrapolate costs and effects of the intervention beyond the trial period to a more appropriate time horizon.”

3) Can you get an access to the appropriate model through the literature review, to input the necessary parameters obtained from the trial?

Yes. As mentioned above, we have previously modelled physical activity to QALYs. We have amended the text accordingly, shown above.

I suggest clarifying the modelling aspect in the long-term analysis

As above, the text on page 24 has been expanded to include further detail.

VERSION 2 – REVIEW

REVIEWER	Sergio Useche University of Valencia, Spain
REVIEW RETURNED	19-Jul-2019
GENERAL COMMENTS	The authors have done a good job addressing most of the comments provided by both reviewers during the last round of

	reviews, However, they failed in adequately strengthen the theoretical-empirical support of the paper (study protocol), limiting the amendments to listing a set of contributions of a single author. Thus, I believe the literature supporting the tool (comment 1 of my past review) is still relatively inadequate, and different issues such as stress, fatigue and shift-work commonly affecting professional drivers (including HGV operators) remain weak. Finally, a further reading proof could be useful to correct many minor writing errors present along the document.
REVIEWER	Inna Feldman Uppsala University, Sweden
REVIEW RETURNED	03-Jul-2019
GENERAL COMMENTS	All my comments were fully addressed.

VERSION 2 – AUTHOR RESPONSE

Reviewer 2

The authors have done a good job addressing most of the comments provided by both reviewers during the last round of reviews, However, they failed in adequately strengthen the theoretical-empirical support of the paper (study protocol), limiting the amendments to listing a set of contributions of a single author.

Thus, I believe the literature supporting the tool (comment 1 of my past review) is still relatively inadequate, and different issues such as stress, fatigue and shift-work commonly affecting professional drivers (including HGV operators) remain weak.

We have moved our citations around slightly in paragraph 1 to provide clearer empirical support for the evidence of the high levels of stress and sleep deprivation seen in HGV drivers. The citation for the recent work of Hege et al. (2018), recommended by Reviewer 2 (included in the originally submitted revised manuscript) now follows the text “...and tight delivery schedules within the Logistics and Transport industry contribute to psychological stress and sleep deprivation,⁶.....”

This reference is also still included at the end of this sentence, along with others, which suggest that stress and sleep deprivation may lead to metabolic disturbances and may further promote unhealthy behaviour choices in HGV drivers.

Further papers on stress in occupational drivers highlighted by the Reviewer in the first round of reviews focused on bus drivers, as our paper concentrates on HGV/truck drivers, we have limited our references to work on HGV/truck drivers only. Furthermore, we do not feel that it would be appropriate to draw too much emphasis to the impact of stress (over and above the other risk factors mentioned, i.e. physical inactivity, smoking, obesity...) as a risk factor in HGV drivers as Hege et al. (2018) observed in their paper that “Stress, however, was not a significant predictor [of CVD risk] in either model and this relationship remains unclear.”

As our intervention primarily targets lifestyle health behaviours such as physical activity, diet and sedentary behaviour, and as unfortunately we are not able to intervene and impact our participants working hours and shift patterns, we feel the level of emphasis on shift work and sleep deprivation is appropriate given the focus of our study. This will certainly be a discussion point however, along with stress, in the outcome papers from this research.

Finally, a further reading proof could be useful to correct many minor writing errors present along the document.

A detailed proofread of the paper has taken place prior to re-submission.

VERSION 3 – REVIEW

REVIEWER	Sergio A. Useche University of Valencia, Spain
REVIEW RETURNED	28-Aug-2019
GENERAL COMMENTS	As the authors have improved the background and support given to the paper by addressing some issues that were remaining, I believe the manuscript can be considered for publication in BMJ Open.